# Optimization of Anthralin Microemulgel Targeted Delivery for Psoriasis and Acne

**DOI:** 10.3390/molecules30122629

**Published:** 2025-06-17

**Authors:** Samiksha Sakarkar, Swati Jagdale, Shrikant Dargude, Anuruddha Chabukswar, Shabana Urooj, Anusha Bilal, Hanan Abdullah Mengash

**Affiliations:** 1Department of Pharmaceutical Sciences, School of Health Sciences and Technology, Dr. Vishwanath Karad MIT World Peace University, Kothrud, Pune 411038, India; sakarkarsamiksha@gmail.com (S.S.); dargudes52@gmail.com (S.D.); anuruddha.chabukswar@mitwpu.edu.in (A.C.); 2Department of Electrical Engineering, College of Engineering, Princess Nourah bint Abdulrahman University, P.O. Box 84428, Riyadh 11671, Saudi Arabia; 3Department of Food Science and Nutrition, University of Leeds, Leeds LS2 9JT, UK; anushabilal24@gmail.com; 4Department of Information Systems, College of Computer and Information Sciences, Princess Nourah bint Abdulrahman University, P.O. Box 84428, Riyadh 11671, Saudi Arabia; hamengash@pnu.edu.sa

**Keywords:** anthralin, psoriasis, acne, microemulgel, central-composite design, molecular docking

## Abstract

**Background:** Anthralin is known for its efficacy in treating psoriasis and acne, possessing poor solubility. Addressing these limitations, the present study endeavors to develop a microemulgel formulation of anthralin aimed at enhancing solubility. **Method:** The solubility study was performed in various solvents. An o/w (oil-in-water) emulsion was formed using the water titration method, which was optimized by statistical experimental design half-run CCD. The final optimized batch was evaluated for physicochemical and in vitro properties **Result:** The final optimized batch showed a particle size (PS) of 417 nm, −25.2 mV zeta potential (ZP) and pH 5.8, which remained stable upon centrifugation, heating–cooling and freeze–thawing cycle. Furthermore, microemulsion with Carbopol 943 5% *w*/*v* was selected as the gel base for the formation of microemulgel characterized by PS, ZP, pH, and viscosity of 230 nm, −50.6 mV, 6.9 and 14,200 cps, respectively, that ensured it a high enough stability. In silico molecular docking between ligand and protein provides the binding energies validating the interaction. Hence, the in silico study was performed for psoriasis and *P. acne* proteins. An in vitro antibacterial activity study on Propionibacterium revealed a significant efficiency of the formulation and MTT assay using L929 cell line in the presence of the drug-loaded microemulgel indicated an inhibition of growth proving that formulation has anti-psoriatic activity. **Conclusions:** Combination therapy with Clindamycin might improve efficacy while reducing antibiotic resistance risks.

## 1. Introduction

The Global Psoriasis Atlas (www.globalpsoriasisatlas.org, accessed on 18 February 2025) and the World Health Organization’s (www.who.int, accessed on 18 February 2025) global report on psoriasis in addition to the National Psoriasis Foundation (www.psoriasis.org, accessed on 18 February 2025) reports that psoriasis is chronic condition affecting over 60 million people worldwide, with an age of onset of 20–30 and 50–60 years, and with a higher prevalence of 2.50% in Western Europe. In 2019, psoriasis accounted for over 3.5 million disability-adjusted life years (DALYs), with more than 40 million prevalent cases and 4.6 million new cases reported worldwide. Furthermore, it has been estimated that by 2030, age-standardized incidence rates per 1 lac were expected to decrease (53.67, 0.00 to 259.99), while incidence cases per 10,000 may increase (487.36, 423.62 to 551.10) [1], and 81% of countries worldwide lack information on the epidemiology of psoriasis [2].

Acne affects around 85% of adolescent and young adult due to the bacteria *Cutibacterium* (formerly *Propionibacterium acnes* a facultative anaerobic Gram-positive bacterium) and the fungi *Malassezia* colonization in the sebaceous sites of the skin [3]. According to a cross-sectional study of the antibiotic-resistant *P. acnes*, strains have been continuously increasing in Thai acne patients. The most common antibiotic resistance was erythromycin, followed by Clindamycin and tetracycline, respectively [4]. *P. acne*’s strains have specific genetic elements associated with their pathogenity. Hence identification and understanding of their role in health and disease by finding genetic differences is crucial for personalized medicines associated with targeted therapies in order to avoid or minimize antibiotic resistance [5].

Anthralin inhibits leukotriene production and LT B4- omega oxidation by human neutrophils, and down-regulates epidermal growth factor receptors, potentially causing antipsoriatic effects. In vitro studies show it inhibits the monocyte secretion of IL-6, IL-8, and TNF-alpha, and even 10 microM anthralin can activate NF-B [6,7]. In acne treatment, several antibiotics are prescribed by oral or topical routes, such as tetracycline, oxytetracycline, doxycycline, minocycline, lymecycline, erythromycin, azithromycin, trimethoprim, and β-lactams ampicillin/amoxicillin/oxacillin; however, their resistance and adverse effects are cause for concern. On the other hand, benzoyl peroxide, azelaic acid, and salicylic acid were found to have anti-microbial properties [8]. Hence, new treatment options are necessary.

Previously, we have reported microemulsion (ME) and microemulgels for various applications [9,10,11,12]. In this study, we developed and characterized anthralin-loaded microemulsion-based gel (microemulgel) with the objective of dual treatment of psoriasis and acne suitable for topical applications. ME can increase the solubility of anthralin and enhance its penetrability through the skin after topical administration, while the gel base can effectively enhance the retention time on the skin. As microemulgels combine, the benefits of gels and microemulsions to treat skin disorders like acne and psoriasis increase. They have a solid gel matrix for controlled release and a small particle size (PS) that improves penetration through the stratum corneum with easy spreadability [13,14].

## 2. Results

### 2.1. Preparation of ME

#### 2.1.1. Screening of Excipients

Microemulsions are essentially clear systems and determining the solubility of the drug in all the components is necessary. The drug solubility measured in oleic acid, Tween 80, and PEG 400 at room temperature (27 ± 1 °C) was found to be of 44.23 mg/mL, 56.017 mg/mL and 14.066 mg/mL, respectively. While Transcutol^®^ exhibited the highest solubility for anthralin in preliminary screening (Figure 1), we opted to use PEG 400 in the final formulation. PEG is a common carrier for drugs with low molecular weights. It possesses low levels of toxicity, immunogenicity, antigenicity, and solubility in water-based solutions. After being administered to living things, it can be quickly eliminated even though it is not biodegradable. Protein or enzyme structure is unaffected by PEG’s presence in water-based solutions [15]. The study found that P-glycoprotein inhibition reduced the transdermal absorption of some drugs. Topical inhibitors can modify P-glycoprotein’s action, and the skin’s defensive role is diminished in aged individuals due to reduced dermis thickness [16,17]. PEG400 has been considered an inhibitor of P-gp [18]. PEG 400 is utilized as a cosurfactant because the surfactant taken solely, i.e., Tween 80, might fail to produce transient negative interfacial tension as well as a fluid interfacial film throughout ME development. Thus, a system of co-surfactants often comes into play to aid in the framework of ME at lower surfactant amounts by dropping the tension between the layers. Co-surfactants also enhance the mobility of surfactant hydrocarbon tails, enabling elevated oil diffusion. The inclusion of a cosurfactant also reduces the bending stress of the interface, giving rise to a more flexible interfacial layer that can accommodate the various curvatures needed to create ME across an array of compositions [19,20]

#### 2.1.2. Construction of Pseudo-Ternary Phase Diagram

The aqueous phase titration method was used to construct a pseudo-ternary phase diagram to reach the required concentration of oil, surfactant, and co-surfactant. By adjusting the ratios of the oil and Smix components, a pseudo-ternary phase diagram was produced. Using aqueous phase titration, the concentration of Smix was reduced by 10% while the concentration of oil was increased by 90%. To create pseudo-ternary diagrams, Smix and oil (oleic acid) were used. Tween 80 to PEG 400 ratios of 2:1, 1:1, and 1:1.5 were also utilized. As shown in Figure 2, areas associated with S_mix_ ratio 1:1, 1:1.5, and 2:1 batch were found to be 97,736.33 ± 496.42, 86,847 ± 977.42 and 102,191.66 ± 676.90. Based on this, 1:1.5 ratio was exempted owing to the least area and *t*-test revealed a significant difference in area between the 1:1 and 2:1 ratio (*p* = 0.025), indicating that the change in area is statistically non-significant. However, considering the toxicity of surfactants towards skin we selected 1:1 area for further analysis.

### 2.2. Experimental Design for ME

Table 1 shows the dependent variables (DVs) and independent variables (IVs) of design with runs and Table 2 shows ANOVA and fit summary of design.

The model F-values of 31.74 for PS, 5.20 for ZP, and 9.35 for %T indicate that the model is significant. There is a 0.07%, 4.21%, and 1.20% chance that an F-value this large will occur owing to noise for PS, ZP, and %T, respectively. *p*-values less than 0.0500 suggest that model terms are significant, but values more than 0.1000 indicate that they are not. In case of PS, B, AB, AC, BC, A^2^, B^2^ are significant model terms, A^2^ is a significant model term for ZP and B, A^2^, B^2^ are significant model terms for %T. The lack of fit (LOF) F-value of 0.22 for PS, 0.0006 for ZP, and 0.12 for %T implies the Lack of Fit is not significant relative to the pure error, and there is a 66.14%, 99.14%, and 74.56% chance that a LOF F-value this large could occur due to noise, respectively.

The predicted R^2^ 0.8773 of PS, 0.8542 of ZP, and 0.7402 of %T is in reasonable agreement with the adjusted R^2^ 0.9518 for PS, 0.7296 for ZP, and 0.8430 for %T, respectively, i.e., the difference is less than 0.2. Adeq Precision measures the signal-to-noise ratio. A ratio greater than 4 is desirable. Hence, the ratio of 17.656 for PS, 6.203 of ZP, and 7.719 for %T indicates an adequate signal. This model can be used to navigate the design space.

Figure 3(a1) shows perturbation plot for PS, as oil (A) increases PS increases, S_mix_ (B) slope shows change in PS as it moves away from the reference point and the water (C) line is flat, which has a negligible effect on PS. Figure 3(a2) shows a perturbation plot for ZP, as oil (A) increases ZP increases, while S_mix_ (B) has a lesser effect on ZP and water (C) has a negligible effect on ZP. Figure 3(a3) shows a perturbation plot for %T, as oil (A) increases, %T decreases, S_mix_ (B) has moderate effect on %T, and water (C) has negligible effect on %T. Considering this we may need to focus on the concentration of oil (A) and S_mix_ (B) while optimizing the formulation. Also, Figure 3(b1–b3) shows a majority of points falling on the straight line, showing good prediction power.

#### 2.2.1. Analysis of PS

PS = 415.85 − 1.76 A − 19.44 B − 3.53 C + 10.71 AB − 26.19 AC − 21.01 BC + 21.25 A^2^ +13.75 B^2^ + 0.0092 C^2^(1)

Equation (1) represents the effect of main effects, two factor interaction and quadratic effects of variables on PS. As per Equation (1), the main effects A, B, and C exhibit negative coefficients of −1.76, −19.44, and −3.53, respectively, indicating that each of these factors contributes to a reduction in particle size (PS). However, the higher negative coefficient of B (−19.44 coefficient) with (F-value of 48.80 and *p*-value of 0.0009) shows its significant effect on reducing PS. The non-ionic surfactant Tween 80 improves the stability of the emulsion. Because of steric hindrance, it creates a protective coating surrounding oil droplet, keeping them from coalescing and preserving their smaller particle size [21,22]. Positive AB interaction (F-value of 7.41, *p*-value of 0.0417) and its coefficient (+10.71) suggests this combination increases the PS. However, as can be seen in Figure 4(a1) and Figure 5(a1), PS becomes low at higher Smix and moderate oil content. At higher concentrations, these components exhibit an increase in particle size (PS). This may be attributed to the addition of oleic acid to Tween 80, which can lead to the formation of micellar aggregates due to oleic acid’s surfactant-like behavior, as reported in earlier studies [23]. Additionally, the AC (F-value of 44.28, *p*-value of 0.0012) negative coefficient (−26.19), as seen in Figure 4(a2) and Figure 5(a2), shows a steep slope indicating strong interaction between oil and water, as reported previously, with viscosity and particle size distribution being, most likely, two ways by which oleic acid (A) influences emulsion PS. Stable emulsions with smaller droplet sizes are produced when the oil–water interfacial region is better contacted due to increased viscosity. However, an excessive amount of oleic acid might cause droplets to enlarge, which will reduce the homogeneity of the emulsion [24,25]. Similarly, BC interaction (−21.01 coefficient) and (F-value of 28.50, *p*-value of 0.0031) decreases the PS and according to Figure 4(a3) and Figure 5(a3), PS becomes low at high Smix and moderate to high water content because more micelles are produced and active species could be easily dispersed, with a direct consequence in decreasing ME particle size. On the other hand, a lower emulsifier concentration leads to a larger particle size dispersion [26]. Finally, quadratic terms A^2^, B^2^, and C^2^ with positive coefficients of +21.25, +13.75, and +0.0092 imply an increase in PS with the increase in its concentration in the ME system.

#### 2.2.2. Analysis of ZP

ZP = −25.06 + 0.954 A − 0.0353 B + 0.176 C −0.298 AB + 0.189 AC −1.07 BC + 4.11 A^2^ + 1.56 B^2^ + 0.160 C^2^(2)

Equation (2) represents the effect of main effects, two factor interaction, and quadratic effects of variables on ZP. As per Equation (2), main effect A(oil) has a positive effect (+0.954) and as reported previously, oleic acid ionizes in aqueous solutions, increasing oil droplet surface charge due to the formation of negatively charged carboxylate ions, enhancing the negative charge on oil droplets [27], C(water) has a positive effect (+0.176), which increases ZP, and B(S_mix_) has a negative effect (−0.0353), which reduces the ZP since tween 80 and PEG-400 both being non-ionic in nature with no charge on their lipophilic portion, hence it reduces ZP and are used in pharmaceutical formulations, as reported previously [28]. However, these effects are insignificant, as suggested by their F-value of 1.02, *p*-value of 0.3587 for oil, F-value of 0.0014, *p*-value of 0.9716 for S_mix_, and F-value of 0.0350, *p*-value of 0.8589 for water. In Figure 4(b1) and Figure 5(b1) at higher S_mix_ and oil content, ZP is low with clear optimal region at moderate oil and S_mix_. ZP reduces due to the negative effect of AB (−0.298) interaction (F-value of 0.0498, *p*-value of 0.8322) and BC (−1.07) interaction (F-value of 0.6418, *p*-value of 0.4594) owing to the non-ionic nature of tween 80 and PEG-400, stearic stabilization, and ionization of oleic acid in the presence of water. There is a positive effect (+0.189) of AC interaction (F-value of 0.0201, *p*-value of 0.8927), suggesting an increase in ZP due to oil and water interaction. Figure 4(b2) and Figure 5(b2) show at moderate oil and high water, higher ZP is observed. Figure 4(b3) and Figure 5(b3) show a steeper curve at both higher concentrations. Finally, quadratic terms A^2^, B^2^, and C^2^ with positive coefficients of +4.11, +1.56, and +0.160 imply an increase in ZP with the increase in its concentration in the ME system.

#### 2.2.3. Analysis of %T

%T = 79.78 − 4.94 A + 11.31 B −0.353 C + 0.696 AB + 9.76 AC + 9.25 BC −14.50 A^2^ −10.00 B^2^ −0.755 C^2^(3)

Equation (3) represents the effect of main effects, two factor interaction, and quadratic effects of variables on %T. As per Equation (3), the main effect of oil with a −4.94 coefficient (F-value of 2.20, *p*-value of 0.1981) and water with a −0.353 coefficient (F-value of 0.0112, *p*-value of 0.9197) does not significantly reduce the %T because oleic acid and water can decrease an %T by increasing ME turbidity. This is because oleic acid forms micelles or droplets that scatter light, reducing the amount of light that can pass through the emulsion. This increased turbidity is due to the formation of a new phase [29]. While S_mix_ (+11.31) increases the %T significantly (F-value of 11.49, *p*-value of 0.0195). This is because the PS of a ME is influenced by the %T and the interaction of Tween 80 and PEG-400 (S_mix_). As reported previously, Tween 80 and PEG-400 adsorb the emulsion’s surface, decreasing the surface tension and increasing the kinetic stability as they prevent oil droplets from aggregating, but high concentrations can cause PS to increase. Surfactants and co-surfactants form micelles, increasing local osmotic pressure and causing droplets to move closer, causing aggregation. Larger particle sizes result in more non-uniform droplets [30]. All interaction effect has a positive influence on %T +0.696 for AB (F-value of 0.0218, *p*-value of 0.8884), which shows %T increases with an increase in Smix and a decrease in oil (moderately) (Figure 4(c1) and Figure 5(c1)), +9.76 for AC (F-value of 4.28, *p*-value of 0.0934), suggesting lower %T at high oil and water (Figure 4(c2) and Figure 5(c2)). These interaction effects can together be explained by the fact that oleic acid also acts as co-surfactant (HLB-1) [31], with the hydrophilic head of oleic acid contains the carboxylic acid group (-COOH), while the hydrophobic tail contains the hydrocarbon group pubchem, which reduces surface tension between oil and water phases, resulting in smaller, uniform droplets. Tween 80 stabilizes the emulsion by forming a protective layer around oil droplets, preventing coalescence and increasing transmittance. PEG 400 further reduces interfacial tension and enhances oil phase solubility, resulting in a clearer emulsion with higher transmittance [32] and +9.25 for BC (F-value of 3.84, *p*-value of 0.1073) producing highest %T at balanced water (around 30%) and Smix (above 33%), as seen in (Figure 4(c3) and Figure 5(c3)) because as reported previously, Tween-80 reduces transmittance and increases viscosity, while PEG decreases it. Both are surfactants, with Tween acting as a surfactant and PEG as an anionic cosurfactant. Combining them produces smaller, stable particle sizes and increases film flexibility, as surfactants alone cannot reduce surface tension [33]. Lastly, quadratic terms coefficient −14.50 for A^2^, −10.00 for B^2^ and −0.755 for C^2^ reduces %T at higher concentrations.

#### 2.2.4. Optimization of ME

The optimized batch given by software is shown in Figure 6. The batch was prepared and the evaluated. The evaluation parameters were found to be in a good range of predicted values confirming the model’s validity. The evaluation of the optimized batch of ME is given in Section 2.3.

### 2.3. Evaluation of ME

#### 2.3.1. Particle Size Analysis, PDI, and Zeta Potential

The microemulsion exhibits polydispersity index (PDI) 0.387, which with good interpretation is less than 0.5. The preparation’s particle size was determined by particle size analysis to be 417 nm. The final batch’s zeta potential was found to be −25.2 mV, demonstrating the emulsion’s good stability as shown in Figure 7.

#### 2.3.2. Dilution Test, %T, Viscosity, and Ph

Phase separation was not seen while diluting the microemulsion with water. This indicates that the o/w kind of microemulsion was used in the final batch. The microemulsion’s pH was found to be around 5.8, meaning that topical treatment is appropriate for it. The viscosity of microemulsion was found to be 3200 cps. Finally, the %T was found to be more than 90%, which indicates a lower particle size.

#### 2.3.3. Thermodynamic Stability

Centrifugation of the microemulsion was performed using a BIO-LAB centrifuge (Mumbai, India) at 5000–10,000 rpm for 30–40 min at 30 °C. Microemulsion physical stability was proven by the absence of phase separation. Furthermore, the heating–cooling cycle and freeze–thaw cycle showed no visible phase separation of microemulsion indicating a stability of the system imparted by the use of optimal concentration of surfactant and co-surfactant.

### 2.4. Evaluation of Microemulsion-Based Gel

#### 2.4.1. Physical Examination

Carbopol 934 was selected over Carbopol 940 and xanthan gum as Carbopol 934 has lower cross-linking density as compared to 940, resulting in required gel viscosity, making the gel matrix for drug to diffuse through, causing higher drug release. Also, xanthan gum is a natural polymer that can favor microbial growth. Carbopol 934 was selected over Carbopol 940 and xanthan gum based on preliminary formulation trials assessing stability, microbial resistance, spreadability, and extrudability. Xanthan gum, although commonly used as a natural gelling agent, showed visible microbial growth during stability studies, indicating poor preservative efficacy. Additionally, gel formulations prepared with Carbopol 934 demonstrated superior extrudability and spreadability compared to those with other gelling agents, as also reported previously [34]. Formulations ranging from 1% to 7% *w*/*v* of Carbopol 934 content exhibited a gradual transition in appearance, consistency, and texture. At lower percentage (1–2% *w*/*v*), the formulations remained transparent, homogeneous, and smooth, with a thin to gel-like consistency. As the concentration increased (3–7% *w*/*v*), the appearance changed to translucent while maintaining homogeneity. The consistency became progressively gel-like, with the 4% *w*/*v* formulation. At higher concentrations (5–7% *w*/*v*), the formulations retained a uniform gel-like texture, ensuring stability.

#### 2.4.2. PS, ZP, and PDI

The microemulgel shows PDI 0.387, which is below 0.5 with good interpretation. Particle size analysis of the preparation showed the size as 230 nm, as shown in Figure 7. Particle size reduces after the incorporation in gel, as also previously reported by [35]. A higher zeta potential value denotes the presence of electrostatic repulsion between two droplets. Either +30 or −30 mV was chosen as the overall dividing point between stable and unstable emulgel. Particles with absolute values of zeta potential greater than or equal to 30 mV (>+30 mV or <−30 mV) are generally considered stable. The optimized batch’s zeta potential was found to be −50.6 mV, demonstrating the emulgel’s good stability.

#### 2.4.3. pH and Viscosity

In individuals with psoriasis, studies have indicated that the skin pH can be lower than that of healthy skin. One study reported an average pH of 5.2 in psoriatic skin, compared to 5.6 in healthy skin [36]. The percentage ionization of the drug was calculated using the Henderson–Hasselbalch equation, employing a previously reported pKa value of 9.06 [37]. At pH 5.2, which corresponds to psoriatic skin conditions, the ionization was found to be only 0.01%, while at pH 5.6, representing healthy skin, it was slightly higher at 0.03%. These results indicate that the drug remains predominantly in its unionized form at both pH levels, which is favorable for topical absorption and permeability. This shows that drugs will be available for absorption through skin. Thus, anthralin’s absorption through skin varies with pH; lower pH increases absorption. The optimal pH range for topical application is 6–7 to balance efficacy and skin irritation [38]. The pH of the formed microemulgel was found around 6.9 which shows good absorption when applied on skin. Also, the viscosity of microemulgel was found to be 14,200 cps.

#### 2.4.4. DSC of Microemulgel

The first endothermic peak at 101.6 °C suggests melting or dehydration, possibly indicating the presence of bound water or a polymorphic transition. The second peak at 134.8 °C suggests further thermal degradation or structural transformation of the sample. The absence of sharp exothermic peaks suggests no crystallization.

#### 2.4.5. Texture Analysis

Understanding the arrangement and interactions between the formulation‘s component parts is aided by the examination of a semi-solid emulsion system‘s mechanical textural properties. Texture profile analysis (TPA) allows assessing particular mechanical characteristics, for example by subjecting a sample to an external compressional stress and evaluating the formulation‘s ability to produce both reversible and irreversible deformations. TPA can ascertain these mechanical properties. A product‘s hardness (expressed in N or gf), defined as its resistance to deformation, is determined by the initial maximum compression force. How long a gel stays in place after applying it to the skin is one indicator of its hardness, which in turn provides information on the gel’s ease of application. The resulting microemulsion-based gel exhibited 9.68 mm deformation at hardness. This result provides a clear indication of the material’s softness, while also highlighting potential limitations in wear resistance. These insights are crucial for determining the appropriate topical applications. It was discovered that the gel had a smooth and uniform texture, without any appearance variations or irregularities [39].

The sample demonstrates a moderate hardness (64 gf) and significant deformation before reaching its maximum force (9.68 mm deformation), as shown in Figure 8. The material undergoes substantial deformation after reaching its peak force, suggesting soft or semi-solid properties. The absence of an abrupt force drop-off after peak load suggests the material maintains some structural integrity post-compression. A higher hardness value corresponds to a firmer gel with greater structural integrity, while a lower value suggests a softer, more spreadable formulation. The observed hardness ensures ease of application while maintaining an adequate viscosity and adhesion for an effective topical delivery.

#### 2.4.6. In Vitro Diffusion Study

Controlled-release formulations are designed to release the drug at a predetermined rate to maintain a consistent drug concentration over a specified period. For an 8 h study, the cumulative drug release should reflect a gradual release pattern rather than a burst release. Initial drug release (0–1 h) starts slowly to avoid a burst release, which is typical for controlled-release formulations. Early to mid-phase (1–4 h) drug release rate increases steadily, ensuring a consistent therapeutic effect. Late phase (4–8 h) release continues to increase gradually, aiming to reach around 90% and above cumulative release by the end of the 8 h. It was observed that the batch F5 showed controlled drug release from topical formulation as compared to reference product, as shown in Figure 9a. Hence batch F5 was selected as a final batch which contained 5% Carbopol 934 polymer.

#### 2.4.7. In Vitro Drug Release: Egg Membrane and Goat Skin

It was determined how much medication will be released cumulatively during a time period of 8 h from the selected batch of microemulgel (Batch F5–5% *w*/*v* Carbopol 934 polymer containing optimized microemulsion formulation) was calculated Figure 9b,c. The release pattern appears to be gradual and sustained, suggesting a controlled release formulation. Initially, there is a rapid release phase (likely due to surface drug diffusion), followed by a steady increase over time. The total cumulative release is likely higher for the egg membrane compared to the goat skin, indicating that the membrane type significantly affects drug permeability.

#### 2.4.8. Stability of Microemulgel

The three-month stability study of microemulgel showed an increase in PS to 253 nm, reduction in pH of formulation to 6.5 making it slightly acidic with no visible changes in physical stability.

### 2.5. In Vitro Activity

#### 2.5.1. Antipsoriatic Activity

MTT assay for microemulgel was performed to determine the cytotoxic property of anthralin against L929 cell lines. Using samples with different drug loads, the minimum cell inhibition (47.90%) at 25 μg/mL concentration, maximum cell inhibition (60.34%)—to 100 μg/mL concentration of anthralin microemulgel and moderate cell inhibition (52.97%)—to 50 μg/mL. The cytotoxic activity of anthralin-loaded microemulgel was compared to microemulgel containing standard drug 5-fluouracil. The results demonstrated that anthralin microemulgel exhibits antipsoriatic potential.

As shown in Table 3 and Figure 10, both 5-FU and anthralin exhibited a dose-dependent cytotoxic effect, with an increasing % inhibition observed at higher concentrations. 5-FU demonstrated greater inhibition compared to anthralin across all tested concentrations. At 100 µg/mL, 5-FU exhibited 79.00% inhibition, whereas anthralin achieved 60.34% inhibition. The IC_50_ value was determined from the linear regression equation obtained from the % inhibition vs. concentration plot (Figure 10). The lower IC_50_ value of anthralin (36.18 µg/mL) indicates that it requires a lower concentration to achieve 50% inhibition compared to 5-FU (54.14 µg/mL), suggesting a relatively higher cytotoxic potency at lower doses. These results suggest that while 5-FU remains more cytotoxic overall, anthralin shows significant inhibition at lower concentrations, making it a potential candidate for further investigation in targeted cancer therapy.

#### 2.5.2. Antibacterial Activity

For this study propionic bacteria were used. After 24 h of incubation, the inhibitory effect of anthralin containing microemulgel was significant as compared to standard Clindamycin (Table 4). Zone of inhibition (ZoI) used as a measure for comparing bactericidal activity of anthralin microemulgel showed value of about 8.66, 10.66, 13.66 mm drug concentration of 25, 50 and 100 μg/mL, respectively, ZoI against the test organisms. The results demonstrated that anthralin microemulgel exhibits antibacterial potential. The increase in inhibition zone with concentration for both Clindamycin standard and antharalin microemulgel indicates a dose-dependent antibacterial effect.

### 2.6. In Silico Docking

Protein domains selected for docking were based on established ligand-binding or active sites, as reported in structural studies. The ligand used was anthralin, and docking was performed at biologically relevant sites previously identified as critical for therapeutic targeting for psoriasis [40,41,42,43,44,45,46] and acne [47,48,49,50,51,52].

The docking study reveals that psoriasis-related proteins and Propionibacterium acnes proteins exhibit varying binding affinities with different ligands, highlighting their potential as drug targets as shown in (Table 5, Figure 11 and Figure 12). Among psoriasis proteins, NF-Kappa-B1 (PDB ID: 8TQD), Interleukin 12 receptor subunit beta−1 (PDB ID: 6WDP) and DYRK2 (PDB ID: 6HDR) exhibit highly negative binding energies, indicating strong and possibly irreversible ligand interactions. Overall, this analysis provides crucial insights into protein–ligand interactions, aiding the identification of potential drug candidates for psoriasis and acne treatment.

## 3. Discussion

The study successfully optimized an anthralin microemulgel for targeted delivery in psoriasis and acne treatment, addressing anthralin’s poor solubility and stability issues. The formulation demonstrated stable physicochemical properties with a particle size of 230 nm, a zeta potential of −50.6 mV, and a viscosity of 14,200 cps, ensuring effective retention and penetration on the skin. The in vitro antibacterial assay confirmed its efficacy against Propionibacterium acnes, while the MTT assay showed significant antipsoriatic activity. The formulation’s optimized composition, utilizing Carbopol 934 as a gel base, enhanced skin adherence and drug release. The study’s thermodynamic evaluation ensured long-term stability, confirming the microemulgel’s suitability for topical application. The drug release studies using egg membrane and goat skin confirmed sustained drug delivery over eight hours. The molecular docking results supported the formulation’s potential interaction with target proteins, suggesting its therapeutic viability. The formulation’s moderate hardness suggests it will provide good retention on the application site without excessive stiffness, ensuring optimal drug release and patient compliance. Anthralin requires a lower concentration (IC_50_ = 36.18 µg/mL) to reach 50% inhibition, indicating its superior effectiveness by comparison with 5-FU (IC_50_ = 54.14 μg/mL) at lower doses. Further optimization of anthralin microemulgel formulation is needed to enhance antibacterial efficacy. Combination therapy with Clindamycin might improve efficacy while reducing antibiotic resistance risks.

## 4. Materials and Methods

### 4.1. Materials

Anthralin was received as a gift sample from Yarrow chemicals, Mumbai, India. Oleic acid, olive oil, castor oil, Transcutol-P, methanol, PEG 400, PEG 600, tween 80, span 80, and tween 20 was received from SRL, Mumbai, India.

### 4.2. Pre-Formulation Study of Microemulsion

#### 4.2.1. Screening of Co-Surfactant, Surfactant, and Oil

Solubility of anthralin in different components, a variety of oils, including oleic acid, olive oil, and castor oil, co-surfactants, including Transcutol-P, PEG 400, and PEG 600, and surfactants like tween 80, span 80, and tween 20, were examined. Excess drug was dissolved in 1 mL of the aforementioned oils, surfactant, and co-surfactant, and vortexed continuously for 48 h. The samples were centrifuged at 5000–7000 rpm for 15 min after which they were left to rest for 72 h. The supernatant liquid was then diluted with ethanol and used to measure the drug concentration via UV-spectroscopy by means of proper calibration curves (absorbance at 360 nm as a function of anthralin concentration) previously obtained.

#### 4.2.2. Emulsification Study

The pseudo-ternary phase diagram comprising lipid, surfactant, cosurfactant and water were developed using the water titration method. This phase diagram was used to select the optimum ratio of Surfactant:Co-surfactant (S:CoS). Pseudo ternary phase diagrams were constructed for oleic acid, Tween 80, and PEG 400 using the water titration method [35]. Selected surfactant (Tween 80) was mixed with co-surfactant (PEG 400) at different weight ratios, i.e., 1:1, 2:1, and 1:1.5 *w*/*w*. The formation of microemulsion was confirmed by attaining a high transparency measured using an optical analyzer. The pseudo-ternary phase diagrams were mapped with CHEMIX software (version 13) and area values were calculated using ImageJ software (version 1.51p, National Institutes of Health).

#### 4.2.3. Optimization by CCD

The half run Central Composite Design (CCD) model by Design Expert software (version 13.0.5, Stat-Ease, Minneapolis, MN, USA) was utilized in the study to examine the link between independent variables (IVs) and dependent variables (DVs). Table 2 lists the experimental levels, DVs, and IVs based on half CCD factorial models. With a minimum of 15 runs, the model looked at three variables at three different levels. The experimental data’s mean values were fitted using a polynomial equation, and each coefficient term’s significance was assessed using an ANOVA. Finding the regression coefficient (R^2^), predicted R^2^, and adjusted R^2^ allowed for the assessment of the experimental data and model. To evaluate the adequacy of the model, graphs displaying perturbation, actual versus predicted values, 3D contours, and 2D plots were created.

#### 4.2.4. Preparation of Microemulsion

To prepare the microemulsion, the pseudo-ternary phase diagram was used with a drug dose of 1.2%. After the preparation of microemulsion, the microemulsion was evaluated for % transmittance, viscosity, pH, etc.

### 4.3. Evaluation of Microemulsion

#### 4.3.1. Percent Transmittance Measurement:

With pure water, microemulsions were diluted up to 100 times. The formulation’s percent transmittance was measured at a wavelength of 660 nm using a UV-visible spectrophotometer with continuous phase (distilled water) serving as the reference.

#### 4.3.2. PS, PDI, and ZP

A Malvern Zetasizer (Nano ZS 90, Malvern Instruments Ltd., Malvern, UK) was used to determine the Microemulsion’s PS, ZP, and PDI. The measurements of the samples were recorded using indisposable cuvettes. Cuvettes were cleaned with methanol and rinsed with a sample for analysis prior to each experiment [35].

#### 4.3.3. Dilution Test, Viscosity, and pH

This experiment was run to determine the type of microemulsion that developed. The prepared microemulsion was diluted with external/continuous phase water. A Brookfield viscometer (Brookfield instruments, AMETEK Brookfield, Mumbai, India) was used to measure the viscosity of the microemulsion. The microemulsion was put in the sample container and the appropriate spindle (No. 7) was placed perpendicularly into it. The spindle was allowed to spin continuously at its optimal speed. The viscosity of the formulation was tested at room temperature. The pH of the microemulsion was measured using the digital pH meter (Lab India PICO+, Mumbai, India) [35].

#### 4.3.4. Thermodynamic Stability

This characteristic was examined to evaluate the physical stability of the microemulsion. To evaluate the system’s capacity to cream or separate phases, the prepared microemulsion was spun for 10 min at 5000 rpm at room temperature. The produced ME system was subjected to a number of stress condition tests, including cycles of heating (40 °C) and cooling (4 °C), freezing (−21 °C), and thawing (+25 °C), while being stored for 48 h at a predetermined temperature. The Microemulsion’s appearance was visually examined [53].

### 4.4. Preparation of Microemulgel

#### 4.4.1. Selection of Polymer for Preparation of Gel Phase

Polymers such as xanthan gum, Carbopol 934, and Carbopol 940 were screened. Separately dispersed in warm water (70–75 °C) and allowed to soak for 24 h, the polymers were prepared for usage. By combining the polymers with distilled water, different gel concentrations were created.

#### 4.4.2. Preparation of Micro-Emulsion-Based Gel

The Carbopol 934 polymer served as the gelling ingredient in the formation of the microemulsion-based gel. The polymer was submerged in water for 24 h. Polymer screening was conducted based on preliminary formulation trials evaluating stability, microbial resistance, spreadability, and extrudability, leading to the selection of Carbopol 934 over Carbopol 940 and xanthan gum. Then, to create a microemulsion-based gel, the prepared microemulsion was mixed with the gel phase. Finally, the pH was adjusted with trienthanolamine.

### 4.5. Evaluation of Micro-Emulsion-Based Gel

#### 4.5.1. PS, ZP, and pH

Malvern Zetasizer (Nano ZS 90, Malvern Instruments Ltd., Malvern, UK) was used to determine the microemulsion’s PS and ZP using a disposable sizing cuvette at a temperature of 25°C [35]. The pH of microemulsion-based gels in 1% aqueous solutions was determined using a digital pH meter [35].

#### 4.5.2. Texture Analysis and Viscosity

The mechanical textural properties of formulations were analyzed at 25 °C using TexturePro CT V1.3 Build 15. A TA-BT-KT texture analyzer set to TPA mode (Stable Micro Systems, Surrey, UK) was used to examine the texture profiles. To analyze texture profiles, 50 mL falcon tubes containing 10 g gel were centrifuged at 4000 rpm for 10 min. The sample was positioned 10 mm above the analytical probe, which was then lowered 10 mm into the sample after being held at a constant speed of 1 mms¹ by the trigger force for 2 min. After that, it was brought back to the surface at a velocity of 0.5 mms^1^, and after 5 s, a second compression was initiated. The experimental procedure was carried out at 25 °C. The viscosity of the microemulgel was determined by using a Brookfield viscometer (Brookfield instruments, AMETEK Brookfield, Mumbai, India).

#### 4.5.3. In Vitro Drug Release Study

##### In Vitro Diffusion Study: Cellophane Membrane

For the in vitro diffusion study, a modified Franz diffusion cell was employed. The permeation medium used was a cellophane membrane number 12, which has a molecular weight cut-off of 12,000 Dalton, an average pore size of 2.4 nm, and an approximate capacity of 1.61 mL/cm. Prior to use, the cellophane membrane was kept in phosphate buffer (pH 5.5) for a total of 12 h. The cell was divided into donor and receptor compartments, with one gram of gel in the donor compartment and 25 milliliters of phosphate buffer in the receptor compartment. The cell was stirred at 50 rpm and the temperature set at 37 °C. After 15 min for first 1 h then after one hour at intervals of one hour, two hours, three hours, four hours, five hours, six hours, seven hours, and eight hours, 1 mL aliquots were removed and replaced with fresh, warm phosphate buffer pH 5.5 each time. At 360 nm, the absorbance of the anthralin containing sample were measured [54].

##### In Vitro Diffusion Study: Egg Membrane

Similarly to the in vitro diffusion study with cellophane membrane, another test was carried out employing egg membrane. A raw egg was obtained, and its contents were gently removed by carefully poking a small hole in it. The membrane was then collected after being immersed in 0.1 N HCl for an overnight period. A concentrated HCl solution can be applied to remove membrane effectively. Following a cleaning with distilled water and a pH 5.5 phosphate buffer, this fresh egg membrane was employed for investigation. Every single series of measurements was performed by using a newly detached egg membrane. The exact same experimental setup was used for the cellophane membrane investigation.

##### In Vitro Diffusion Study: Goat Skin

The goat’s fresh dorsal skin was donated by a butcher. Skin hair was extracted and preserved in tyrode solution. Cleaned goat skin piece was placed on the epidermis facing the donor compartment. The phosphate-buffer solution (pH 5.5) was then added to the receptor compartment, which was kept at 37 °C ± 0.5 °C using magnetic stirring. The membrane was coated with 1 gm of microemulgel, equivalent to (1.2% anthralin), and a drug release process was conducted like that used with cellophane membranes.

#### 4.5.4. Stability of Microemulgel

After three months of storage at different temperatures (8 °C, 48 °C + 75% RH, and room temperature (RT), the stability of the microemulgel formulation was evaluated. Samples were examined for organoleptic changes, physical stability, pH, and PS [55].

#### 4.5.5. Antipsoriatic Activity

The L929 mouse connective tissue cell line was received from the National Center for Cell Sciences (NCCS), Pune, India and grown in DMEM media containing 10% fetal bovine serum. The cells were kept at 37 °C in a 5% CO_2_ incubator and seeded at a density of 1 × 10⁴ cells/mL in culture media for 24 h. To conduct the experiment, 70 μL of cell suspension (1 × 10^4^ cells/well) were added to 100 μL of culture media on a 96-well tissue culture plate. Sample was added to wells at varying doses (25, 50, and 100 μg/mL), while control wells contained 0.2% DMSO in PBS with cell culture. All experiments were performed in triplicate to ensure reliability and reproducibility of the results. Control wells were utilized to evaluate cell viability and calculate the percentage of cells that survived after incubation. The cultures were incubated at 37 °C with 5% CO_2_ for 24 h in a CO_2_ incubator (Thermo Scientific BB150, Thermo Fisher Scientific Inc., Waltham, MA, USA). After incubation, the media were removed and 20 μL of MTT reagent (5 mg/mL in PBS) was added. The plates were incubated for a further 4 h at 37 °C to allow live cells to convert the MTT into formazan crystals, which were then viewed under a microscope. After removing the medium, 200 μL of DMSO was added and incubated at 37 °C for 10 min while protected from light with aluminum foil. Each sample’s absorbance was measured at 570 nm using an ELISA microplate reader (Benesphera E21, Avantor Performance Materials India Private Limited, Gurgoan, India), with triplicates tested for accuracy [56].

#### 4.5.6. Determination of Antibacterial Activity

The antimicrobial activity was performed to determine anti-acne activity of anthralin containing microemulgel by pour plate method. Nutrient Agar (NA-Himedia, HiMedia Laboratories Pvt. Ltd., Mumbai, India) was prepared by dissolving 28.0 g in 1000 mL distilled water, sterilized at 121 °C for 15 min, and poured into sterile Petri plates. **Propionibacterium** was used for microbial assays. A 24 h pure culture was prepared by suspending the microorganisms in saline solution, streaking onto culture slants, and incubating at 37 °C. Standard solutions, including Clindamycin (25 mg/mL), were prepared in sterile water and stored under refrigeration. Plates were prepared using the pour plate method by pouring melted agar into sterile plates and allowing its solidification before use.

A sterile, non-toxic cotton swab was dipped in the standardized inoculums (turbidity was adjusted to achieve confluent growth on the Petri plate), and the entire plate’s agar surface was streaked with the swab three times, turning the plate at a 60-degree angle between streaking. The streaked inoculums were then allowed to dry for 5–15 min with the lid on. Then, using sterile well punches, a bore was punched on the prepared plates (8 mm). With the assistance of a sterile micropipette, a 10 μL dosage of produced microemulsion and standard medication Clindamycin were injected into each bore in an aseptic environment. Plates were maintained at room temperature for 30 min before being incubated for 24 h at 37 °C. The diameter of the inhibitory zones was measured using a scale in mm [57].

The antibacterial potential of test compounds was assessed using the mean diameter of the zone of inhibition around the disc in millimeters. The zones of inhibition of the studied microorganisms by the substances were determined on a millimeter scale.

#### 4.5.7. In Silico Molecular Docking

In silico molecular docking was performed using BIOVIA Discovery studio (Discovery Studio 2021) on various proteins of psoriasis and *P. acne* bacteria. Briefly, the ligand was minimized and docked with minimized and prepared proteins. The docking score, 2D and 3D images are shown [58].

## 5. Conclusions

The optimized anthralin microemulgel provides a promising solution for psoriasis and acne treatment, enhancing drug solubility, stability, and skin penetration. The formulation exhibited effective antibacterial and antipsoriatic properties. Its controlled-release mechanism ensures prolonged therapeutic action. Combination therapy strategies may further enhance treatment outcomes. These findings suggest microemulgels as an alternative for improving topical drug delivery in dermatological applications.

## Figures and Tables

**Figure 1 molecules-30-02629-f001:**
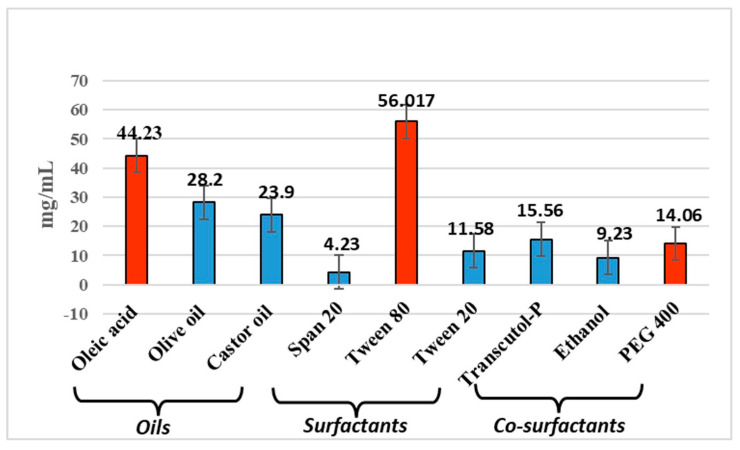
Screening of excipients.

**Figure 2 molecules-30-02629-f002:**
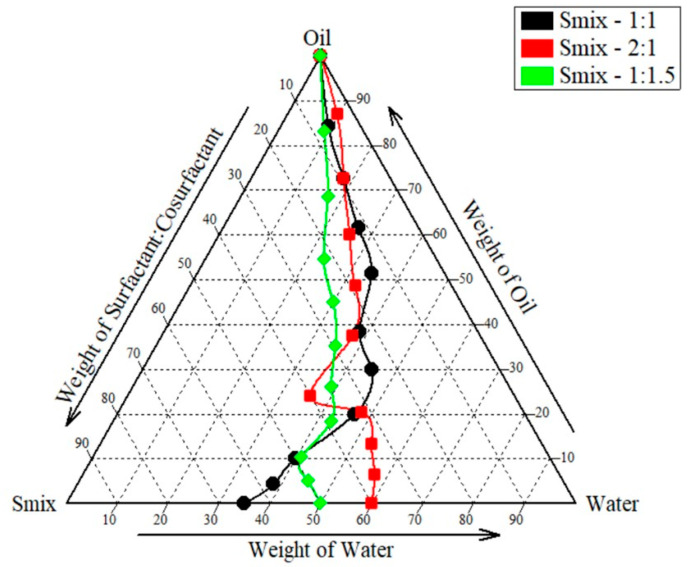
Pseudo-ternary phase diagram.

**Figure 3 molecules-30-02629-f003:**
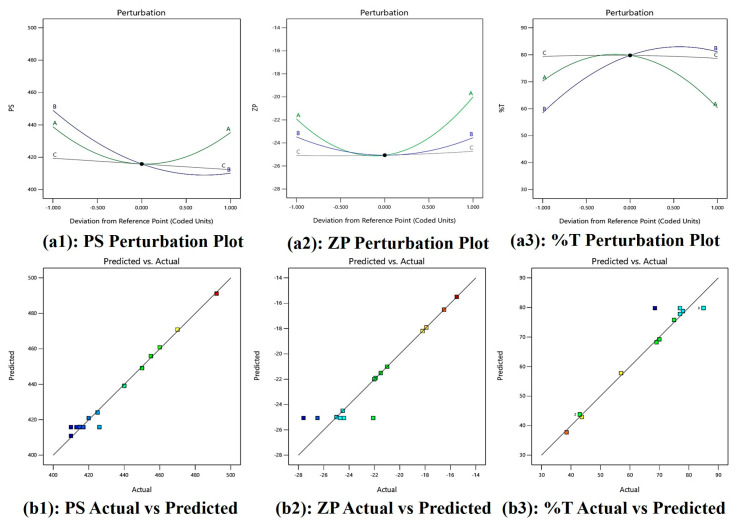
(**a1**–**a3**) Perturbation plots and (**b1**–**b3**) actual vs. predicted plot of PS, ZP, and %T.

**Figure 4 molecules-30-02629-f004:**
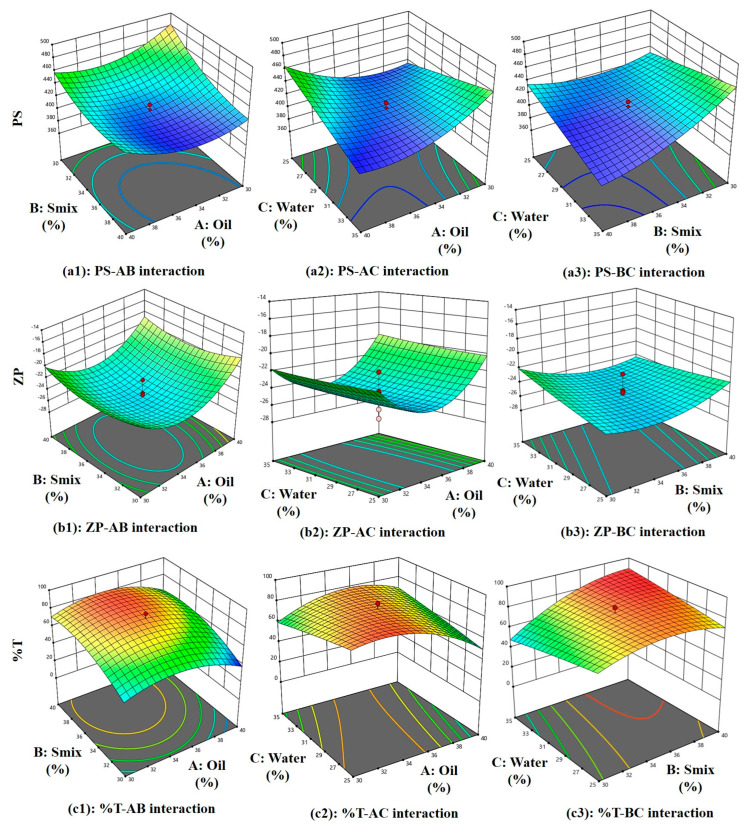
Three-dimensional surface plots of the PS, ZP, and %T.

**Figure 5 molecules-30-02629-f005:**
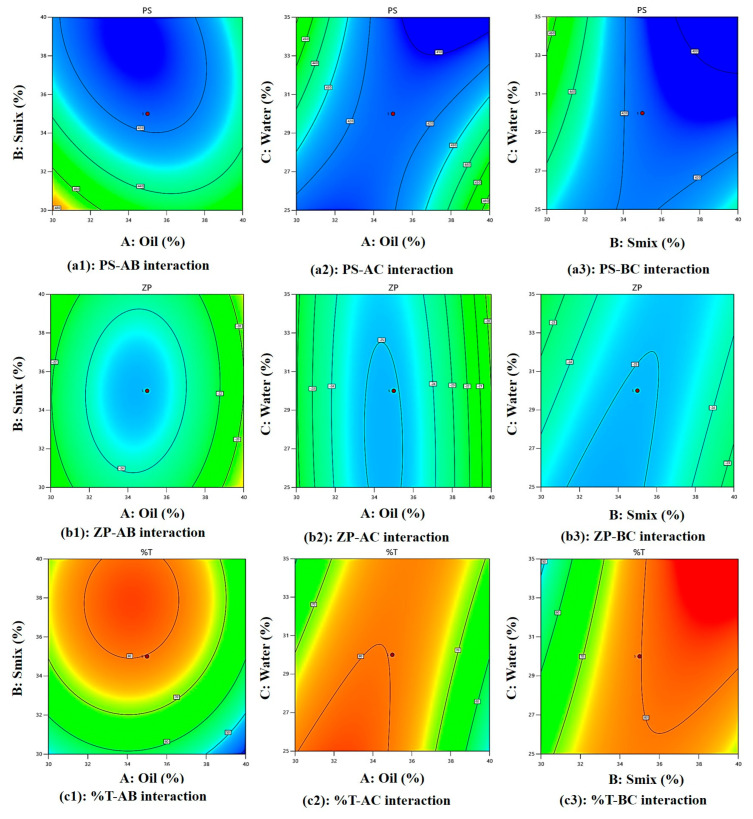
Two-dimensional contour plots of the PS, ZP, and %T.

**Figure 6 molecules-30-02629-f006:**
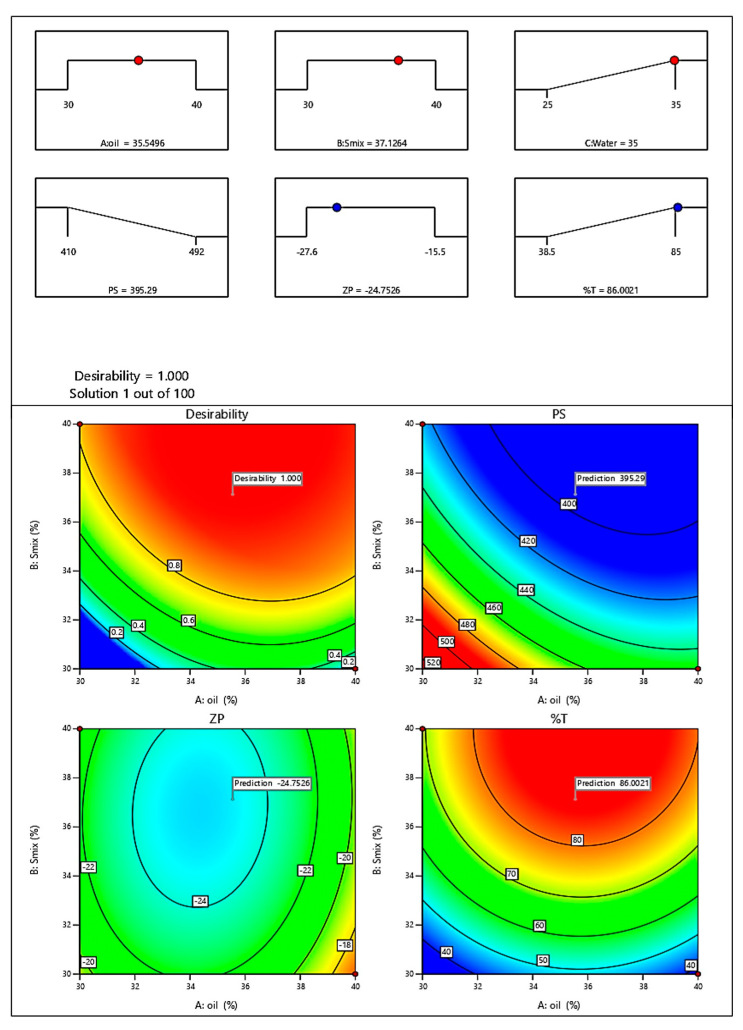
Optimized batch of ME.

**Figure 7 molecules-30-02629-f007:**
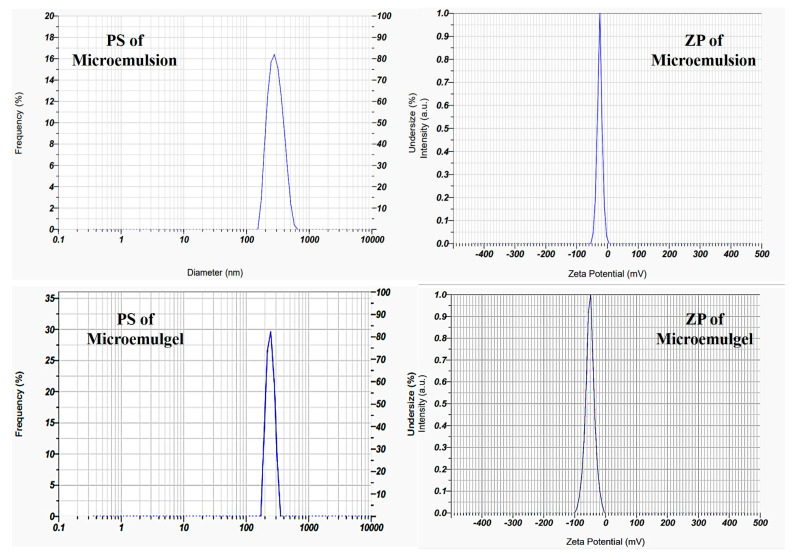
Particle size and zeta potential of optimized batch of ME and microemulgel.

**Figure 8 molecules-30-02629-f008:**
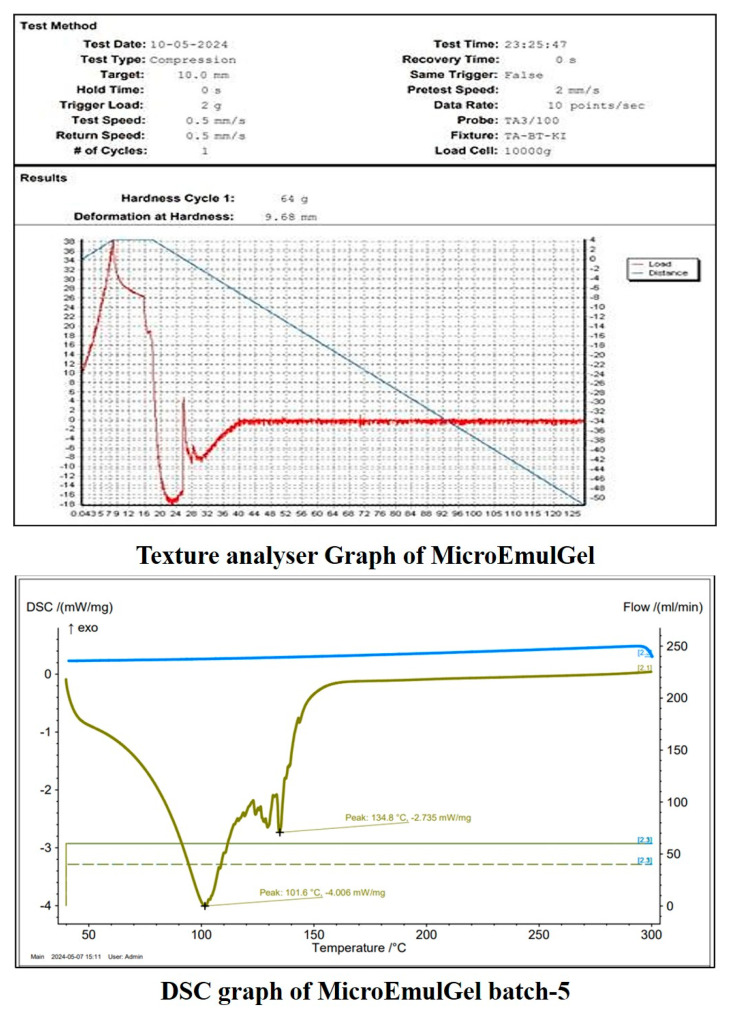
Evaluation of microemulgel batch.

**Figure 9 molecules-30-02629-f009:**
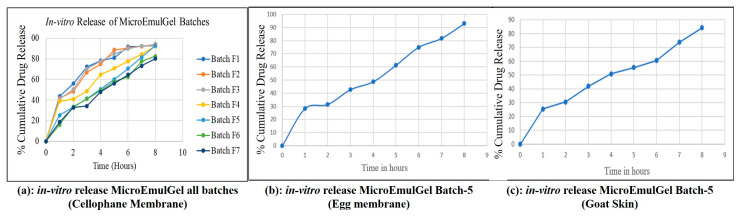
In vitro release profiles of microemulgel batch.

**Figure 10 molecules-30-02629-f010:**
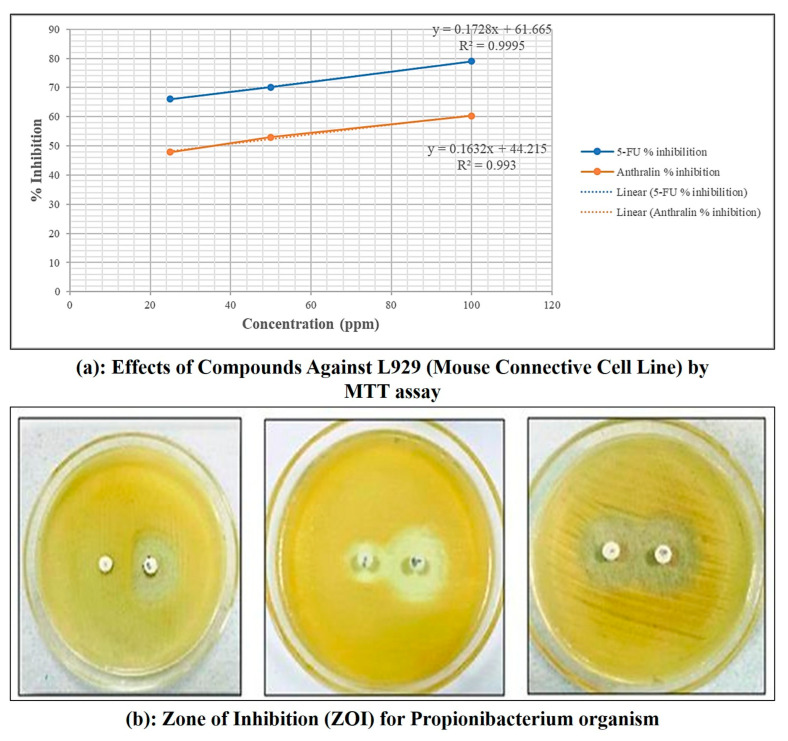
MTT assay and zone of inhibition for Propionibacterium organism of microemulgel batch.

**Figure 11 molecules-30-02629-f011:**
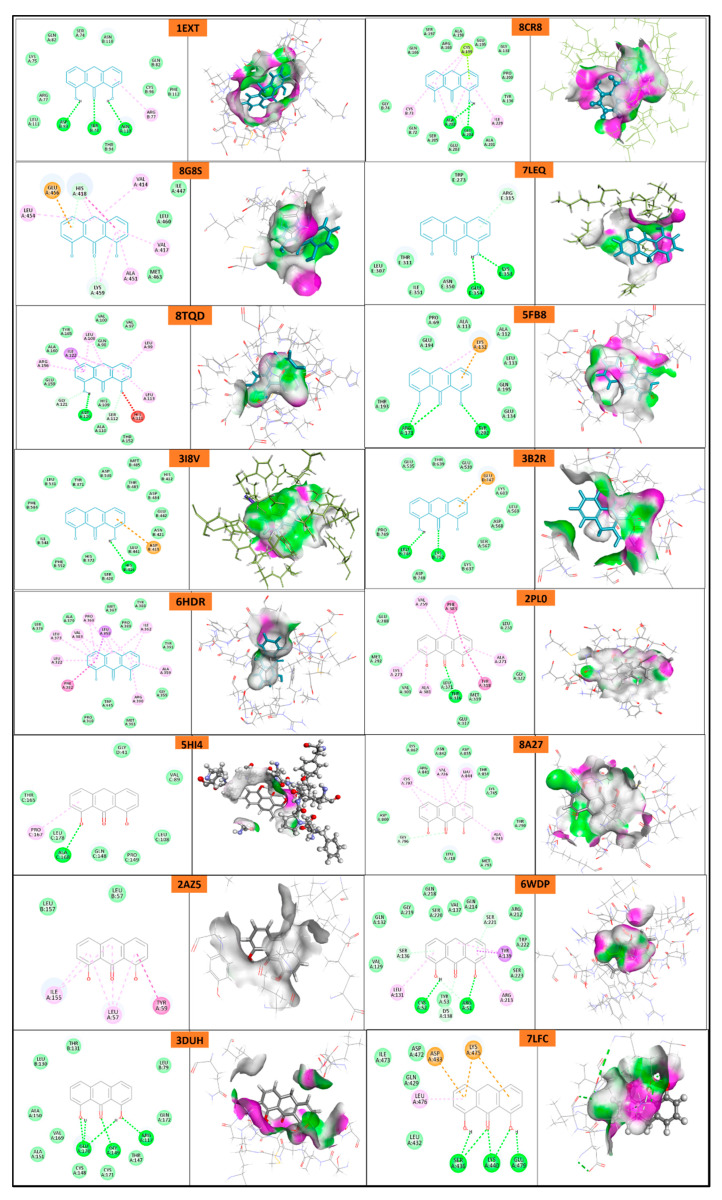
Molecular docking of psoriasis proteins with anthralin.

**Figure 12 molecules-30-02629-f012:**
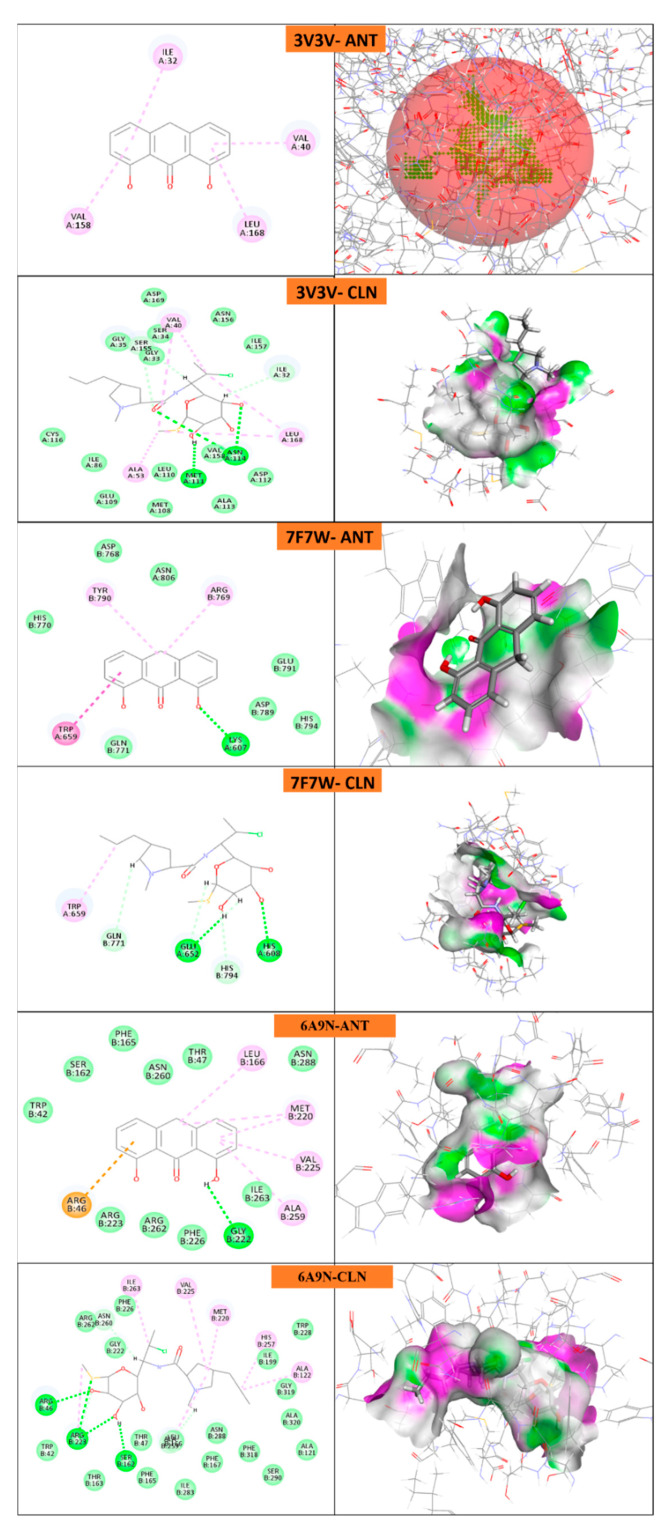
Molecular docking of *P. acne* proteins with anthralin (ANT) and Clindamycin (CLN).

**Table 1 molecules-30-02629-t001:** Selection of dependent variables (DVs), independent variables (IVs), and half-run central composite design (CCD) runs and results.

Design—Half Run CCD										
Independent Variables (IVs)			Levels				Dependent Variables (DVs)			
			−1		0	+1				
**1**	Oil		30		35	40	**1**	Particle Size (PS)		
**2**	S_mix_		30		35	40	**2**	Zeta Particle (ZP)		
**3**	Water		25		30	35	**3**	%Transmittance (%T)		
**Experimental Runs and Results**										
**Batch**	**Oil (%)**	**Smix (%)**		**Water (%)**			**PS (nm)**		**ZP (mV)**	**%T**
**F1**	35	35		30			417		−24.7	85
**F2**	35	35		30			413		−24.4	77
**F3**	35	35		37.0711			410		−24.5	77
**F4**	35	35		22.9289			420		−25	78
**F5**	40	30		35			450		−16.5	38.5
**F6**	30	40		35			425		−21	70
**F7**	35	35		30			426		−27.6	68.5
**F8**	30	30		25			440		−21.5	69
**F9**	35	42.0711		30			415		−22	75
**F10**	40	40		25			492		−17.9	43.7
**F11**	42.0711	35		30			455		−15.5	43
**F12**	27.9289	35		30			460		−18.2	57
**F13**	35	27.9289		30			470		−21.9	43
**F14**	35	35		30			410		−26.5	85
**F15**	35	35		30			415		−22.1	85

**Table 2 molecules-30-02629-t002:** Summary of ANOVA for PS, ZP, and %T.

ANOVA for the Responses (PS, ZP, and %T)													
Source	Sum of Squares			df	Mean Square			F-Value			*p*-Value		
	PS	ZP	%T		PS	ZP	%T	PS	ZP	%T	PS	ZP	%T
**Model**	8854.75	167.04	3749.58	9	983.86	18.56	416.62	31.74	5.20	9.35	0.0007	0.0421	0.0120
**A-Oil**	12.50	3.65	98.00	1	12.50	3.65	98.00	0.4033	1.02	2.20	0.5533	0.3587	0.1981
**B-S_mix_**	1512.50	0.0050	512.00	1	1512.50	0.0050	512.00	48.80	0.0014	11.49	0.0009	0.9716	0.0195
**C-Water**	50.00	0.1250	0.5000	1	50.00	0.1250	0.5000	1.61	0.0350	0.0112	0.2600	0.8589	0.9197
**AB**	229.60	0.1779	0.9701	1	229.60	0.1779	0.9701	7.41	0.0498	0.0218	0.0417	0.8322	0.8884
**AC**	1372.40	0.0719	190.66	1	1372.40	0.0719	190.66	44.28	0.0201	4.28	0.0012	0.8927	0.0934
**BC**	883.49	2.29	171.13	1	883.49	2.29	171.13	28.50	0.6418	3.84	0.0031	0.4594	0.1073
**A^2^**	3486.52	130.32	1623.17	1	3486.52	130.32	1623.17	112.48	36.50	36.44	0.0001	0.0018	0.0018
**B^2^**	1460.45	18.78	772.29	1	1460.45	18.78	772.29	47.12	5.26	17.34	0.0010	0.0704	0.0088
**C^2^**	0.0007	0.1979	4.40	1	0.0007	0.1979	4.40	0.0000	0.0554	0.0989	0.9965	0.8232	0.7659
**Residual**	154.98	17.85	222.73	5	31.00	3.57	44.55						
**Lack of Fit**	8.18	0.0006	6.53	1	8.18	0.0006	6.53	0.2229	0.0001	0.1209	0.6614	0.9914	0.7456
**Pure Error**	146.80	17.85	216.20	4	36.70	4.46	54.05						
**Cor Total**	9009.73	184.90	3972.32	14									
**Fit Statistics**													
		**PS**	**ZP**		**%T**			**PS**		**ZP**		**%T**	
**Std. Dev.**		5.57	1.89		6.67	**R^2^**		0.9828		0.9034		0.9439	
**Mean**		434.53	−21.95		66.31	**Adjusted R^2^**		0.9518		0.7296		0.8430	
**C.V. %**		1.28	8.61		10.06	**Predicted R^2^**		0.8773		0.8542		0.7402	
						**Adeq Precision**		17.6557		6.2031		7.7193	

**Table 3 molecules-30-02629-t003:** Effects of active compounds against L929 (Mouse Connective Cell Line) by MTT assay.

Sr. No.	Conc (µg/mL)	Absorbance (OD)				Cell Viability %	Inhibition %	IC_50_ (µg/mL)
		1	2	3	Mean			
Reference Standard—5-flurouracil (5-FU)								
5-FU	25	0.48	0.376	0.34	0.398	33.90	66.10	54.14
	50	0.333	0.325	0.396	0.351	29.87	70.13	
	100	0.292	0.241	0.208	0.247	21.00	79.00	
Experimental Drug—Anthralin								
Anthralin	25	0.615	0.623	0.6	0.6126	52.097	47.902	36.18
	50	0.548	0.547	0.564	0.553	47.023	52.976	
	100	0.475	0.466	0.458	0.4663	39.654	60.345	

**Table 4 molecules-30-02629-t004:** Comparative data of antibacterial activity of Clindamycin (standard) and anthralin loaded in microemulgel.

Sr. No.	Concentration (μg/mL)	Zone of Inhibition (mm) (Propionibacterium)	
		(Clindamycin) Standard	(Anthralin Microemulgel)
**1**	25	11.33 ± 1.15	8.66 ± 1.52
**2**	50	13.66 ± 1.15	10.66 ± 1.52
**3**	100	20.66 ± 1.15	13.66 ± 1.15

**Table 5 molecules-30-02629-t005:** In silico docking study.

Protein Name	PDB ID	Binding Energy (Kcal/mol)	Interacting Ligand at Binding Site	
			Bonding Type	Binding Amino Acid
**1. Docking with Psoriasis proteins**				
**Tumor Necrosis Factor Receptor**	1EXT	18.87	H-bond	ASP B: 93 ASN A: 110 SER B: 94
			Pi-alkyl	ARG B:77
**Human Interleukin-23**	8CR8	−102.71	H-bond	ALA A:202 GLU A:204
			Pi-lone pair	CYS A:199
			Alkyl	CYS B:73 ILE A:229
**p52**	8G8S	14.86	Pi-anion	GLU A:456
			Alkyl	LEU A:454 VAL A:417 ALA A:451
			Pi-Pi T-shaped	HIS A: 418
			Van der waals	HIS A:418 LYS A:459
**p50**	7LEQ	11.08	H-Bond	GLU E:354 LYS E:353
			Pi-H Bond	ARG E:315
**NF-Kappa-B1**	8TQD	−970.66	H-bond	ASP A:120
			Pi-sigma	ILE A:122
			Alkyl	LEU A:113 LEU A:99 LEU A:108 ARG A:156
**Interleukin-16**	5FB8	19.54	H-bond	ARG A:171 TYR A:202
			Alkyl and Pi-cation	LYS A:132
**PDE4a**	3I8V	20.31	H-Bond	HIS B:416
			Pi-Anion	ASP B:413
**PDE5A1**	3B2R	19.82	H-Bond	LEU B:746 LYS B:752
			Pi-Anion	GLU B:747
**DYRK2**	6HDR	−788.57	Pi-sigma	LEU A:358
			Pi-Pi T-shaped	PHE A:382
			Alkyl	LEU A:322 LEU A:373 VAL A:383 PRO A:368 ILE A:362 ALA A:359 ARG A:390
**LCK**	2PL0	19.95	H-bond	THR A:316
			Pi-Pi stacked	PHE A:383 TYR A:318
			Alkyl	VAL A:259 ALA A:273 LYS A:318 ALA A:271
**IL-17A**	5HI4	11.12	H-bond	ALA C:168
			Pi-alkyl	PRO C:167
**EGFR kinase**	8A27	−8.99	Pi-alkyl	CYS A:797 VAL A:726 LEU A:844 ALA A:743
			C-H bond	GLY A:796
**Vitamin D nuclear receptor ligand binding domain**	2HBH	22.26	H-Bond	LEU A:258
			Pi-Pi T-shaped	HIS A: 333
			Pi-alkyl	VAL A:328 ILE A:299
**TNF-alpha**	2AZ5	6.34	Alkyl	LEU A:57 ILE A:155
			Pi-Pi stack	TYR A:59
**Interleukin 12**	6WDP	−107.23	H-Bond	CYS A:52 ARG A:51
			Pi-Sigma	TYR A:139
			Pi-alkyl	LEU A:131 ARG A:213 TYR A:139
			Van der waal	ARG A:51 TYR A:53
**Structure of Interleukin-23**	3DUH	17.36	H-bond	GLU A:170 GLY A:149 ARG A:117
**Structure of importin a3 bound to p50 NLS**	7LFC	−19.3	H-bond	SER A:436 LYS A:440 GLU A:479
			Pi-alkyl	LEU A: 476
			Pi-anion/Pi-cation	ASP A:433 LYS A:475
**2. Docking with *P. acne* proteins**				
**JNK1**	3V3V-ANT	24.60	Pi-alkyl	ILE A:32 VAL A:158 LEU A:168 VAL A:40
	3V3V-CLN	−22.73	H-Bond	ASN A:114 MET A:111
			Alkyl	VAL A:40 LEU A:168 ALA A:53
			C-H Bond	ILE A:32 SER A:155
**JAK2-JH2**	7F7W-ANT	19.99	H-Bond	LYS A:607
			Pi-Pi T-shaped	TRP A:659
			Alkyl	TYR B:790 ARG B:769
	7F7W-CLN	0.355	H-bond	GLU U:652 HIS A:608
			Pi alkyl	TRP A:659
			C-H Bond	HIS B:794 GLN B:771
**KAS III from Propionibacterium acnes**	6A9N-ANT	11.89	H-bond	GLY B:222
			Alkyl	ALA B:259 VAL B:225 MET B:220 LEU B:166
			Pi-cation	ARG B:46
	6A9N-CLN	−37.36	H-bond	ARG B:46 ARG B:223 SER B:162
			Alkyl	ILE B:263 VAL B:225 MET B:220 HIS B:257 ALA B:122

## Data Availability

Data will be made available from corresponding author on request.

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
