# Peer review of "Optimization of Anthralin Microemulgel Targeted Delivery for Psoriasis and Acne"

_molecules, 2025, doi:10.3390/molecules30122629_

Round 1

Reviewer 1 Report

Comments and Suggestions for Authors

Some suggestions/comments are inserted into the attachment.

Comments on the Quality of English Language

Written English should be generally reformulated in clearer, shorter and meaningful sentences/phrases.

Author Response

Please find the attached file Response to Reviewer 1 Round 1.

Reviewer 2 Report

Comments and Suggestions for Authors

The manuscript entitled “Optimization of Anthralin Microemulgel Targeted Delivery for Psoriasis and Acne” describes the preparation, characterization and in vitro evaluation of O/W microemulsions loading anthralin and incorporated in Carbopol gels for the topical treatment of psoriasis and acne.

Although the development of formulations to improve the efficacy of topical therapies in the management of acne of psoriasis is an interesting field of research, the manuscript under review shows too many inaccuracies and discrepancies that makes its scientific soundness questionable.

First, English is poor making the meaning of many sentences unclear. For instance, at line 38, the meaning of the sentence “In 2019, there were over 35 lac disability-adjusted life year (DALY) cases, over 4 Cr prevalence cases, and over 46 lac incidence cases of psoriasis worldwide” is unclear. At line 145, the meaning of the sentence “As per eq. 1, ,ain effects A, B, and C with a negative coefficient of -1.76, -19.44 and – 3.53 145 show these effects reduce the PS” is unclear. At line 153, the meaning of the sentence “While higher concentration of these implies an increase in PS showing its quadratic effect this is due to the addition of oleic acid to Tween 80 increases particle size, possibly due to OA's role as a surfactant, causing the formation of micellar aggregates, reported previously” is unclear. These are inly few examples of sentences whose meaning is unclear. English should be carefully revised to allow an accurate comprehension of the text.

A few examples of inaccuracies and discrepancies I found in this manuscript are listed below.

  1. In Fig. 1, the authors show that Transcutol was the best solvent for anthralin but they chose PEG 400 as solvent without providing any explanation of this choice.
  2. In Table 1, the sum of the percentages of each component used to prepare microemulsions is not 100%. In some cases, values are reported using four digits after the decimal point. Values should be reported using the same digits after the decimal point.
  3. In Table 2, in some cases, values are not reported using the same digits after the decimal point.
  4. At line 249, the authors report “The microemulsion's pH was found to be between 5.8”. What is the meaning of “between 5.8”?
  5. At line 283, it is unclear why ME particle size was 230 nm after incorporation in gel formulations while it was 450 nm in the initial preparation.
  6. At line 148, it is unclear what proteins the authors are referring to.
  7. At line 348, what is the meaning of CDR?
  8. At line 354, what is the composition of batch E?
  9. At line 269, the author report they selected Carbopol 934 but in the material and methods section they did not describe what tests they carried out to perform this selection.
  10. The experimental conditions used to perform PS, ZP, viscosity, in vitro release studies are not clearly described.
  11. At line 538, the authors cite Fig. 3.5 to illustrate the modified Franz-cells they used but there is not a Fig. 3.5 in the manuscript.
  12. At line 549, the authors cite Fig. 3.7 but there is not a Fig. 3.7 in the manuscript.
  13. The discussion is only seven lines.
  14. The authors do not provide any conclusion.
Comments on the Quality of English Language

The quality of English is very poor. English should be carefully revised throughout the manuscript.

Author Response

Please find the attached file Response to Reviewer 2 Round 1.

Round 2

Reviewer 1 Report

Comments and Suggestions for Authors

The manuscript was changed according to the suggestions made on the old version of the paper.

Reviewer 2 Report

Comments and Suggestions for Authors

The authors addressed most criticisms properly but English needs to be revised.

Comments on the Quality of English Language

The authors provided a revised version of the manuscript but the quality of English language should be improved.